# SARS-CoV-2 transmission dynamics in Belarus in 2020 revealed by genomic and incidence data analysis

Alina Nemira [1], Ayotomiwa Ezekiel Adeniyi [1], Elena L. Gasich[2], Kirill Y. Bulda[2], Leonid N. Valentovich [3], Anatoly G. Krasko[2], Olga Glebova[1], Alexander Kirpich[4,5] & Pavel Skums [1,5✉]

## Abstract

**Background** Non-pharmaceutical interventions (NPIs) have been implemented worldwide to curb COVID-19 spread. Belarus is a rare case of a country with a relatively modern healthcare system, where highly limited NPIs have been enacted. Thus, investigation of Belarusian COVID-19 dynamics is essential for the local and global assessment of the impact of NPI strategies.

**Methods** We integrate genomic epidemiology and surveillance methods to investigate the spread of SARS-CoV-2 in Belarus in 2020. We utilize phylodynamics, phylogeography, and probabilistic bias inference to study the virus import and export routes, the dynamics of the effective reproduction number, and the incidence of SARS-CoV-2 infection.

**Results** Here we show that the estimated cumulative number of infections by June 2020 exceeds the confirmed case number by a factor of ~4 (95% confidence interval (2; 9)). Intra-country SARS-CoV-2 genomic diversity originates from at least 18 introductions from different regions, with a high proportion of regional transmissions. Phylodynamic analysis indicates a moderate reduction of the effective reproductive number after the introduction of limited NPIs, but its magnitude is lower than for developed countries with large-scale NPIs. On the other hand, the effective reproduction number estimate is comparable with that for the neighboring Ukraine, where NPIs were broader.

**Conclusions** The example of Belarus demonstrates how countries with relatively low outward population mobility continue to be integral parts of the global epidemiological environment. Comparison of the effective reproduction number dynamics for Belarus and other countries reveals the effect of different NPI strategies but also emphasizes the role of regional Eastern European sociodemographic factors in the virus spread.

### Plain language summary

Belarus is one of few European countries that has enacted limited measures to contain SARS-CoV-2, the virus that causes COVID-19. We study the genetic sequences of the SARS-CoV-2 virus circulating in Belarus and other countries in 2020 to investigate how it might have been imported into the country and spread there. We show that the virus was repeatedly imported from and exported to different regions, including a large portion of regional transmissions that occurred despite stricter measures implemented by Belarus' neighbors. There was a moderate reduction of the virus reproductive number—a measure of virus transmission speed—after April 2020, but its magnitude was lower than for developed countries with more stringent epidemiological interventions. These findings shed light on the COVID-19 spread in Eastern Europe and highlight the impact of public health policies and of regional factors on this spread.

[1] Department of Computer Science, Georgia State University, Atlanta, GA, USA. [2] Republican Research and Practical Center for Epidemiology and Microbiology, Minsk, Belarus. [3] Institute of Microbiology, National Academy of Sciences of Belarus, Minsk, Belarus. [4] Department of Population Health Sciences, School of Public Health, Georgia State University, Atlanta, GA, USA. [5]These authors contributed equally: Alexander Kirpich, Pavel Skums.
✉email: pskums@gsu.edu

The Republic of Belarus is a country in Eastern Europe with a population of approximately 9.5 million. In comparison to other ex-USSR and Central European countries, it is characterized by weaker socio-economic and political ties with the neighboring European Union[1,2] and lower outward population mobility[3]. At the same time, Belarus has a relatively modern healthcare system[4], and the country's Human Development Index (HDI) has been categorized as "very high" (vhHDI)[5].

The COVID-19 epidemic has reached Belarus later than most Western European countries and approximately at the same time as its neighbors. The first confirmed imported case reported on February 28, 2020, was a person who arrived from Iran[6,7]. Since then, there was a steady increase in the number of officially reported laboratory-confirmed cases that has surpassed 300,000 on March 13, 2021.

The major feature of the COVID-19 pandemic in Belarus is the notably narrower scope of non-pharmaceutical interventions (NPIs) in comparison to other vhHDI countries[8]. The implemented NPIs included a mandatory 14-day self-isolation for individuals who were arriving from abroad or were identified as close contacts of individuals with confirmed COVID-19; some social distancing measures such as the increase in the frequency of public transportation operations to reduce crowding; remote teaching and delaying the class starting times at schools and higher education institutions[9]. No large-scale quarantines, lockdowns, or other strict social distancing measures have ever been administered. The other widely practiced measures such as mask regimen and border closures were not mandated until November and December of 2020, respectively (Fig. S1).

Given the uniqueness of the Belarusian experience, understanding COVID-19 epidemiological dynamics in this country is essential not only for the assessment of its past and current public health situation but also for a better insight into the impact of different NPI strategies around the globe. However, the development of such understanding has been impeded by the limited amount of available data. Until the last quarter of 2020, the only available data have been the officially reported country-level counts that included daily incidence, numbers of conducted diagnostic tests, and COVID-related mortality. Such statistics are prone to biases and underreporting[10,11]. While these drawbacks are well-known and common for all countries, they have the potential to be exacerbated in Belarus due to limited testing capacities provided by a handful of national-level laboratories[9].

In the meantime, whole-genome sequencing (WGS) data analyzed using genomic epidemiology methods provides a complementary and independent source of information. WGS SARS-CoV-2 data have already been used to study transmission histories and epidemiological dynamics in a variety of countries and administrative regions[12–20]. For Belarus sufficiently representative genomic dataset has become available only in late 2020, when the limited sequencing data produced outside of the country on behalf of the World Health Organization (WHO) were extended by the locally produced sequences.

In this paper, we combined WGS genomic data and epidemiological data to carry out the first study of SARS-CoV-2 transmission dynamics in Belarus. In the absence of large amounts of reliable epidemiological statistics, the integrated genomic and incidence analysis allowed to fill the information gap and provide a plausible picture of the emergence and spread of SARS-CoV-2 in the country. The obtained results also gave insight into the effect of limited NPIs during the first epidemic wave. Specifically, the analysis revealed a large number of unobserved infections, multiple virus introductions from different regions, and a considerable amount of virus importations between Belarus and its geographic neighbors. Furthermore, the epidemiological dynamics during the first epidemic wave in

Belarus were comparable to those in the neighboring Ukraine, where NPIs were broader. We also were able to observe a moderate reduction of the basic reproduction number after the introduction of limited NPIs, which, however, was lower in magnitude in comparison to the countries with large-scale NPIs.

## Methods

**Data**. The SARS-CoV-2 genomic data for analysis were downloaded from GISAID[21] on March 15, 2021. The Belarusian dataset consists of 41 full-length genomes sampled between March 2020 and February 2021. Specimen and their metadata were originally collected by the Republican Research and Practical Center for Epidemiology and Microbiology (RRPCEM, 40 specimens) and Gomel State Medical University (1 specimen) in surveillance settings. The sequenced samples were selected to represent all 7 major regions of Belarus; however, 60% of sequences represent the capital city of Minsk, where the testing facilities were better developed and whose population constitutes 22% of the country's total population. Females constitute 51% of sampled individuals, the age distribution was: ≤20 years–15%, 21−40 years–26%, 41−60 years–49%, and ≥61 years–10%. One sequence was obtained from a citizen of Azerbaijan who was tested in Belarus to be allowed to return to his home country. This sequence is marked as Azerbaijanian in GISAID but is considered as Belarusian here. The Ukrainian dataset that was analyzed for comparison purposes consists of 116 sequences. Daily numbers of new cases and conducted tests were collected from the official Telegram channel of the Ministry of Health of the Republic of Belarus[22].

**Global phylogenetic analysis**. For the phylogeny reconstruction, we utilized the SARS-CoV-2-specific phylogenetic inference pipeline implemented in Nextstrain[23]. The sequences from Belarus were analyzed together with 12,064 background sequences from the global SARS-CoV-2 population. To obtain a representative sample with those background sequences, a country-specific Nextstrain context subsampling was used[23]. The sequences were aligned using MAFFT[24], and a maximum likelihood (ML) phylogenetic tree was constructed using IQ-TREE[25] under Hasegawa-Kishino-Yano (HKY)+Γ nucleotide substitution model with a gamma-distributed site rate variation[26].

In the resulting time-labeled tree, ancestral geolocation traits have been inferred using the so-called "mugration model"[27]. In this model, countries of origin of the tree nodes are considered as discrete traits, and the virus spread between countries is considered as a general time-reversible process. We augmented this model by incorporating the human mobility statistics provided by European Commission Knowledge Center on Migration and Demography (KCMD)[28] via KCMD Dynamic Data Hub[3]. Even though global travel has been affected by COVID-19-related restrictions, these statistics are still assumed to representatively reflect the relative density of human mobility between countries even in quarantine settings. Specifically, the transition rates between traits were assumed to be proportional to the normalized average numbers of inter-country trips. The resulting transition rate matrix has been used to estimate the maximum joint likelihood traits of internal nodes using the dynamic programming algorithm[29]. This trait inference algorithm has been implemented in Matlab (v. R2019b).

Belarusian clades were defined as those having the most recent common ancestors (MRCA) with "Belarus" trait, and intra-Belarusian lineages were inferred as the maximal subtrees inside these clades. Upon examination of the Belarusian clades, we joined two clusters that have the same estimated source trait and the MRCA at the tree distance of 4 from both of them. Finally,

global lineages of sequences were determined using Pangolin SARS-CoV-2 Lineage Assigner[30].

**Intra-country phylodynamic analysis**. In this work, we largely followed a general analytic pipeline adopted in other similar country-level studies (see e.g., refs. [12,13,15,31]), with several modifications tailored for the specifics of the analyzed data. At first, the temporal signal was evaluated by constructing an ML phylogeny under HKY+Γ nucleotide substitution model and by regressing root-to-tip genetic divergence against sampling dates using TempEst (v.1.5.3)[32]. Next, BEAST (v.2.6.3)[33] was used to fit the Coalescent Bayesian Skyline model to the full set of Belarusian sequences. As before, HKY+Γ nucleotide substitution model was used together with a strict molecular clock. The clock rate was assumed to follow a gamma (Γ) distribution with the mean equal to $8 \times 10^{-4}$ mutations/site/year and the standard deviation of $5 \times 10^{-4}$ [13,34], where the distribution density was parametrized using the corresponding shape and rate parameters. Four segments were assumed for the effective population size that roughly corresponded to growth and decline periods of the first and second COVID-19 epidemic waves. The model parameters were sampled from the corresponding posterior distribution using Markov Chain Monte Carlo (MCMC) method with $3 \times 10^7$ iterations, sampling every $3 \times 10^3$ iterations and the initial 10% "burn-in" iterations. The MCMC sampling quality was assessed using Tracer (v.1.7.1)[35] and accepted if all parameters had effective sampling sizes (ESS) higher than 200. The obtained maximum clade credibility (MCC) tree was annotated using Tree Annotator (v.1.8.4)[36]. The reliability of intra-Belarusian clusters detected by the ML phylogenetic inference was re-confirmed by verifying their correspondence to monophyletic clades in the MCC tree. For each cluster, time to the most recent common ancestor (TMRCA) was estimated.

The effective reproduction number $\mathcal{R}_e$ and the sampling proportion have been estimated for the two best-sampled Belarusian transmission lineages with a total of 19 genomes (Table S3) using the Birth-Death Skyline Serial (BDSKY) model[37] implemented in BEAST. The analyzed lineages were likely co-circulating over the same susceptible population (see Results). Thus, we used a linked model where both lineages evolve and are being sampled independently but share the substitution model parameters, the molecular clock rate, and the effective reproduction number drawn from the same respective priors. Given the relative sparsity of available genomic data, this approach allows using larger and more representative combined samples for the analysis. The same settings as above have been used for the substitution model, molecular clock, and MCMC. Since the BDSKY model is parameter-rich, we equipped it with informative priors on several parameters. Specifically, the sampling proportions were assumed to have a *Beta* ($\alpha,\beta$) distribution prior with parameters $\alpha = 1$ and $\beta = 9.99 \cdot 10^5$, thus reflecting the sparsity of the Belarusian sequence sample (the proportion of sequenced cases from the total number of cases is assumed to vary between $10^{-6}$ and $10^{-3}$). The prior for the origin of each cluster was assumed to be normally distributed with the mean equal to the time estimated using the Coalescent Bayesian Skyline. For the rate of becoming non-infectious, we assumed an infectious period of 10 days[12,13,38,39]. Finally, we considered the models with one and two changes of the effective reproduction number $\mathcal{R}_e$ and the sampling proportion. The times of the parameters change were fixed to July 1, 2020, for the first model and May 1 and July 1, 2020, for the second model. The list of model parameters is reported in Table S1.

**Inference of case counts**. Here we used two complementary approaches. In the first approach, trees and BDSKY parameters sampled by BEAST were used to reconstruct cumulative case count trajectories using the particle filter algorithm implemented in EpiInf (v.7.3.0)[40]. In the second approach, we utilized the method of ref. [11]. It quantifies the case counts underestimation from the numbers of confirmed cases and conducted tests up to a specified date in a semi-Bayesian way under the assumptions that the observed data are subject to sampling, reporting, and diagnosis biases. The model[11] has been used with the default settings. The case count trajectories were inferred by taking $10^4$ samples from model-defined prior distributions of testing probabilities for individuals with different severity of symptoms.

## Results

**SARS-CoV-2 genomic diversity**. In total, the observed Belarusian SARS-CoV-2 41 sequences constitute 0.02% of officially reported cases. Despite such sparse sampling, the observed Belarusian SARS-CoV-2 sequences belong to 11 genomic lineages (by the nomenclature of ref. [41], Fig. 1a). In particular, the genome that was sampled on February 23, 2021, belongs to B.1.1.7 lineage that emerged in the UK in November 2020 and had been rapidly spreading toward fixation[42]. The root-to-tip regression analysis demonstrated a moderately strong temporal signal ($R^2 = 0.62$, $p = 1.03 \cdot 10^{-9}$, Fig. 1c).

**SARS-CoV-2 transmission history**. We identified 18 distinct intra-Belarusian clades that most likely correspond to separate introductions of SARS-CoV-2 into the country. The inference of between-country importations of SARS-CoV-2 is usually complicated since during the global pandemic close genomic variants can be observed in multiple geographic locations. Therefore, the results of such inference should be treated with caution. With that in mind, we note that the inferred transmission history agreed with the travel records for those cases when they were available. In particular, the first confirmed SARS-CoV-2 case was the individual who arrived from Iran[7], and the phylogenetics reaffirmed that. The agreement also held for the second detected case brought by the traveled from Italy[43]. The first introduction produced at least one secondary case as indicated by the tree; however, both lineages were not sampled after March 2020 (Fig. 2b). This can be attributed to the timely isolation of those individuals and their first order contacts[44]. In general, SARS-CoV-2 importations into the country could be attributed to a mixture of regional and global transmissions. As illustrated by Figs. 1b and 2c, the most frequent alleged virus introduction sources were the neighboring countries of Russia (5 introductions) and Poland (3 introductions).

Five SARS-CoV-2 introductions (28%) are associated with clusters of two or more sequences and thus are hypothesized to establish intra-country transmission lineages. The three largest transmission lineages are paraphyletic and may indicate virus re-export from Belarus to other countries. Even though some alleged export cases could be sampling artifacts, those of them involving large lineages are more reliable. Such cases include two SARS-CoV-2 introductions to the neighboring country of Latvia in June 2020 (95% CI: May 31, 2020–June 26, 2020) and in October 2020 (95% CI: October 1, 2020–October 22, 2020) that established substantial Latvian transmission lineages (Fig. S3)

**Effective reproduction number and the effect of NPIs**. Most observed clades originated between March and July of 2020, and the majority of their times to MRCA fall into April (Figs. 1d, 2b, Fig. S2 and Table S4). The majority of branching events also belonged to the same time period. It implies that despite sequencing being performed mostly in late 2020—early 2021, the phylodynamic analysis of currently available Belarusian genomes

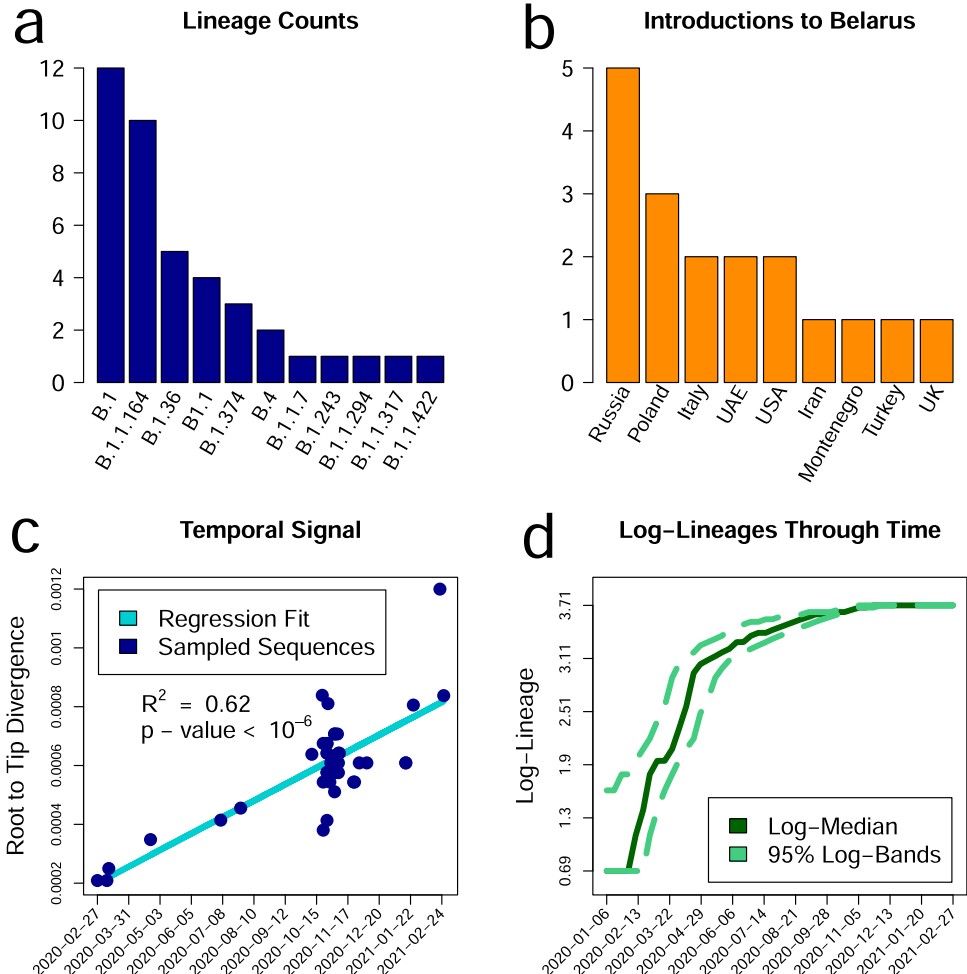

**Fig. 1 Summaries of Belarusian SARS-CoV-2 genomic diversity and intra-Belarusian transmission lineages produced by the phylogenetic analysis. a** Numbers of sequences (*Y*-axis) in 11 genomic lineages present in Belarus (*X*-axis). **b** Sources of virus introductions to Belarus. *X*-axis: estimated countries of origin of Belarusian lineages. *Y*-axis: number of lineages that originated in each country. **c** Temporal signal assessed by linear regression for sequence sampling times (*X*-axis) versus root to tip divergence in the ML phylogeny (*Y*-axis). Belarusian sequences are depicted as blue dots, the linear regression line is highlighted in cyan. **d** Lineages-through-time (on a logarithmic scale). *X*-axis: calendar time. *Y*-axis: log-number of lineages in the viral phylogeny with living descendants at each time point. Log-Median and 95% confidence intervals for each time point reflect the uncertainty of lineage birth times estimation.

allows us to reliably assess only the first epidemic wave prior to July 2020. Another reason to choose July, 1 for the endpoint of our phylodynamics analysis is the dynamics of the daily percentage of positive tests. The WHO criterion for influenza-like illnesses (ILI) assumes that the epidemic is "under control" if the percentage of positive tests is below 5% for at least two weeks[45]. According to the officially reported data, Belarus reached this state with respect to the first COVID-19 wave by the end of June 2020 (Fig. 3d and Fig. S1), even though the reported incidence peaked several weeks earlier.

Best-sampled transmission clusters are well-mixed and have representatives from at least two Belarusian administrative regions (Fig. 2b). This fact and the relative homogeneity of the Belarusian demographical characteristics suggest that the corresponding viral lineages co-circulated over the same susceptible population. Thus, we estimated the effective reproduction number $\mathcal{R}_e$ for these lineages using a linked Birth-Death Skyline (BDSKY) model. The model with three segments shows a moderate decline of the median $\hat{\mathcal{R}}_e$ from $\hat{\mathcal{R}}_e = 1.95$ (95% highest posterior density (HPD) interval: (1.03; 2.99)) in March-April to $\hat{\mathcal{R}}_e = 1.59$ (95% HPD interval: (0.82; 2.39)) in May–June

(Fig. 4b). The obtained HPD intervals, however, are rather wide due to the relatively small genome sample size. Thus, we also estimated the median $\hat{\mathcal{R}}_e$ for the entire period of March-June, which turned out to be $\hat{\mathcal{R}}_e = 1.70$ (95% HPD interval: (1.45; 1.96)). The Kolmogorov-Smirnov test was used for the formal comparison of prior and posterior distribution samples for $\hat{\mathcal{R}}_e$ and resulted in $p < 10^{-10}$ for all of them.

In addition, we matched the estimate of the effective reproduction number $\hat{\mathcal{R}}_e$ for Belarus against that for Ukraine—the neighboring ex-USSR non-EU country with similar demographics. The major difference in COVID-19 epidemics between Belarus and Ukraine is the scope of NPIs, with Ukraine implementing much stricter lockdown and physical distancing policies[8]. The same Birth-Death Skyline Serial model was applied to two best-sampled Ukrainian clusters with a total of 28 sequences defined as in ref. [46] (Table S3). The median Ukrainian $\hat{\mathcal{R}}_e$ over the same time period was estimated to be $\hat{\mathcal{R}}_e = 1.64$ (95% HPD interval: (1.49; 1.81)). This assessment agrees with the previous estimation based on the Exponential Coalescent model[46] and appeared to be comparable to $\hat{\mathcal{R}}_e$ estimates for Belarus.

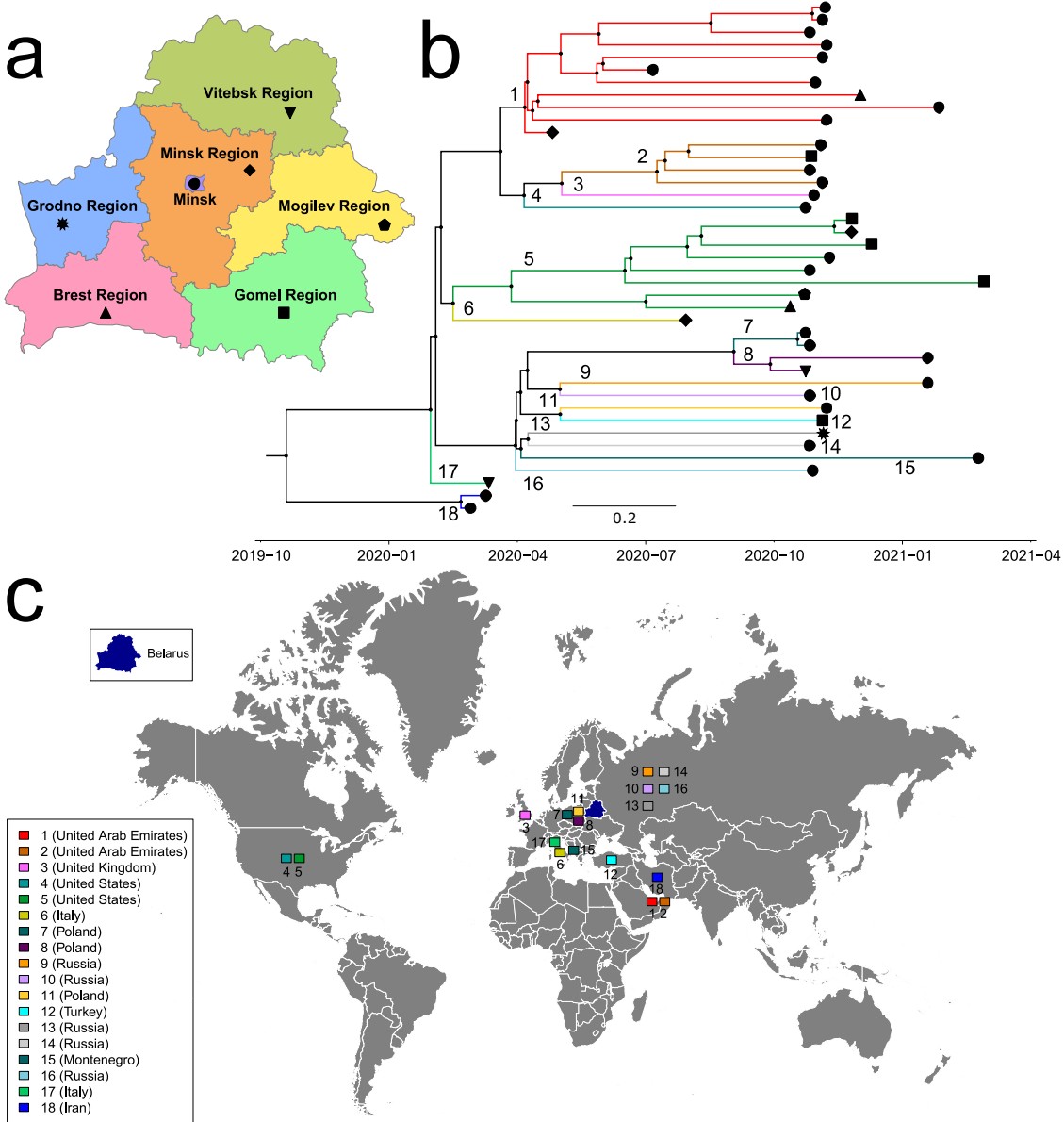

**Fig. 2 The annotated maximum clade credibility tree of Belarusian sequences constructed using the Coalescent Bayesian Skyline model. a** Administrative regions of Belarus. Each region is marked by a particular black shape. **b** The annotated tree. The tree leaves are shape-coded by sampling regions. The tree branches that form each of 18 intra-country transmission lineages are highlighted by a particular color. The root of each lineage is also marked by the lineage id (an integer between 1 and 18). The table of lineage ids, color codes, and estimated countries of origin is depicted at the lower-left corner of the figure. One can see that the largest lineages are observed in multiple geographic regions. **c** Origins of transmission lineages marked on the world map by their ids and colors. The numbers of lineages that originated in each country are also shown in Fig. 1b. One can see that a large portion of lineages originated from the country's neighbors. Belarus and world outline maps by Vemaps.com

**Cumulative incidence and case counts**. Cumulative case count trajectories for Belarus implied by the BDSKY model are reported in Fig. 4c. The cumulative number of cases by July 1, 2020, falls into the 95% prediction interval: (364; 17066). It should be kept in mind that these estimates apply only to two transmission lineages out of possibly many more.

The results of the complementary analysis based on the number of officially reported cases $D(t)$ and conducted tests $T(t)$ over time are presented in Fig. 3. The model[11] was designed for the initial phase of the epidemics with the exponential growth of $D(t)$. Therefore, we calibrated and used it to estimate the cumulative number of infections $C(t)$ for the time interval from April 1 (the first date when the number of conducted tests was available) to May 16, 2020 (officially

reported peak of the first wave) with a 15-day increment (Fig. S1). The obtained results suggested a substantial underestimation of the cumulative number of cases through the study period (Fig. 3b). In particular, on $t^* =$ May 16, 2020, the model predicted $C(t^*) = 118,521$ cases (95% PI: (54,057; 249,000)) while the reported number was $D(t^*) = 28,681$. Hence, 76% of infections that occurred by that date were supposedly undetected (95% PI: (47%; 88%)). The model-inferred case detection rate $D(t)/C(t)$ increases over time as more tests are being conducted (Fig. 3c).

## Discussion

In this paper, we presented the first detailed study of the COVID-19 epidemic in Belarus using the officially reported incidence data,

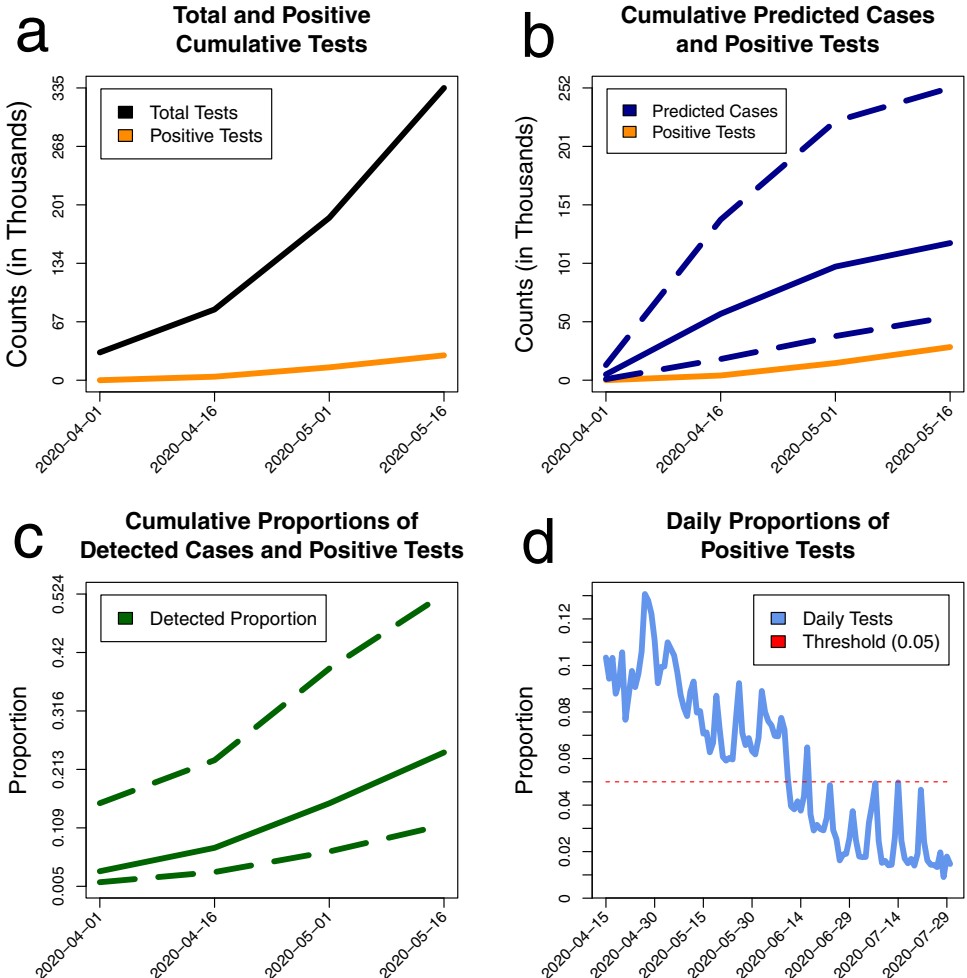

**Fig. 3 The summary of counts data analysis using the model of ref. [11]. a** The plots of input data: officially reported cumulative numbers of cases $D(t)$ (orange) and conducted tests $T(t)$ (black). **b** The cumulative numbers of officially reported cases $D(t)$ (orange) plotted together with the case counts $C(t)$ that were inferred by the model (blue). The solid blue line represents median estimates across $10^4$ model runs, while dashed lines depict 2.5th and 97.5th percentiles. The figure suggests a substantial underestimation of the true case counts by the observed data. For both **a** and **b**, X-axis represents a calendar time, and the Y-axis—counts of cases and tests. **c** Model-based case detection rate over time $t$ calculated as $D(t)/C(t)$. As in **b**, solid and dashed lines depict the median and 2.5th and 97.5th percentiles, respectively. **d** The daily proportion of positive tests (light blue). The Red dashed line highlights the WHO-suggested threshold of 0.05: the proportion of positive tests should be below this threshold for at least two weeks for the epidemic to be considered "under control".

testing data, and genomic data collected between March 2020 and February 2021. The reported results substantially expand our understanding of COVID-19 dynamics and the effects of limited NPIs in Belarus and reflect several key epidemiological issues that it shares with other countries around the globe.

First, the analysis revealed the diverse history of transmissions of SARS-CoV-2 into, from, and inside the country. It identified 18 introductions within 11 genomic lineages, which is comparable with similar early estimates for other regions[12,14,15,18,20]. However, for countries with denser sequence sampling, these numbers are considerably larger[13,16,17]; thus the estimate for Belarus is most likely a lower bound on the real number of introductions. Still, it allows for several important observations. In contrast to most Western European and North American countries[13–20,39], the larger portion of estimated transmission links was with geographic neighbors. It is not entirely surprising, given the comparatively lower outward mobility of the Belarusian population. It is also worth mentioning that much stricter travel restrictions implemented by Belarus' neighbors failed to stop the flow of SARS-CoV-2 across the borders in both directions. Furthermore, approximately half of the estimated introductions did not appear

directly across the border, which emphasizes that Belarus, like most countries in the world, is a part of a global interconnected environment and as such, affects and is affected by epidemiological developments in other countries.

Second, the estimation of the effective reproduction number $\mathcal{R}_e$ allowed the preliminary assessment of the effect of limited NPIs implemented in the country during the first epidemic wave. These estimates should be interpreted only in comparison with similar estimates for other countries. The analysis suggests a moderate but statistically significant decrease of $\mathcal{R}_e$ after the NPIs were put in action (Fig. 4b). The magnitude of decrease, however, is lower in comparison to the countries with broader and stricter NPIs (Table S2). Furthermore, the estimated median effective reproduction number $\hat{\mathcal{R}}_e = 1.70$ (CI: (1.45; 1.96)) over the entire analyzed period for Belarus is comparable with the estimates of $\mathcal{R}_e$ in developed countries before the introduction of strict NPIs[15,16,39,47]. For example, for Victoria, Australia this value is 1.63 (CI: (1.45; 1.8))[15]. On the other hand, the estimate of $\mathcal{R}_e$ for Belarus is also close to the estimate of $\mathcal{R}_e$ over the same time period for neighboring Ukraine, where the scope of implemented NPIs has been much broader. In our opinion, the latter fact is not

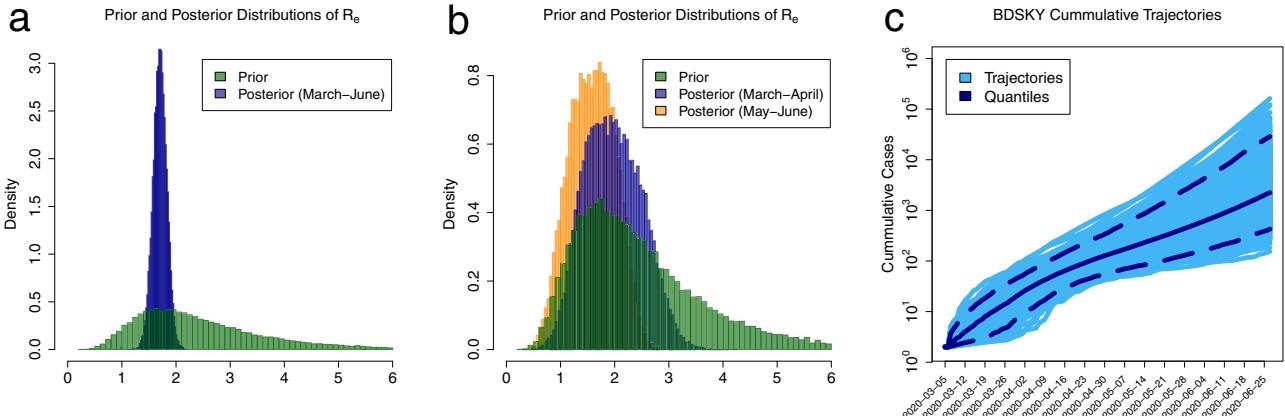

**Fig. 4 BDSKY model estimations of the effective reproduction number $\hat{\mathcal{R}}_e$ and cumulative case count for the two largest Belarusian lineages. a** Prior (green) and posterior (blue) distributions of $\hat{\mathcal{R}}_e$ for Belarus in March-June, 2020. **b** Prior (green) and posterior (blue and orange) distributions of $\hat{\mathcal{R}}_e$ for Belarus before and after the introduction of limited NPIs, respectively. For both **a** and **b**, X-axes are the values of $\hat{\mathcal{R}}_e$, and Y-axes are probability densities. **c** Possible cumulative case count trajectories on the scale. X-axis: calendar time. Y-axis: estimated cumulative case counts. Each trajectory is depicted as a light blue line, and overlapping trajectories jointly form a blue-shaded area of the plot. Solid blue and dashed lines represent a median and 95% confidence intervals for case counts at each time point.

entirely surprising and is more reflective of the reported extensive violations of lockdown and distancing measures in Ukraine and the limited ability of authorities to control the epidemics[48,49]. A similar estimate $\mathcal{R}_e = 1.76\,(0.91;\,2.71)$ has been also reported for Russia[12] which borders both Belarus and Ukraine. This comparison of three ex-USSR countries suggests that regional demographic and social specifics could be important factors for COVID-19 epidemiology along with NPIs. The study of such factors should be the subject of further investigation.

Third, the true number of infections by the end of May 2020 is most likely ~4 (CI :(2;9)) times higher than the detected number of cases, which is expected for respiratory diseases in general and for COVID-19 in particular[11,50]. For example, according to the observed seroprevalence of SARS-CoV-2 antibodies, in the USA the total number of COVID-19 infections in March-May, 2020 was probably between 6 to 24 times the number of reported cases[10].

It is important to highlight that the presented study has several limitations. The first of them is the scarcity of currently available genomic data, especially in comparison with most other European countries. Our approach strives to compensate for it by utilizing informative priors and linked models for phylogenetic and phylodynamics inference. BDSKY models are also sensitive enough and suitable for inference even for smaller genomic datasets. For example, the numbers of sequences and/or density of branching events in this study are similar to those in other studies[37,39,51], where meaningful estimates of $\mathcal{R}_e$ have been produced for several epidemics, including SARS-CoV-2. Nevertheless, the inference precision could have been higher, if more SARS-CoV-2 genomes have been available. Furthermore, new data may allow comparing first and second COVID-19 waves, which is especially interesting for understanding the epidemiological effects of social and public health developments during the second half of 2020. In our opinion, however, the lack of other studies justifies the need to fill the knowledge gap and to report the results based on the existing data. We also hope that this study will serve as a trigger for further SARS-CoV-2 genomic epidemiology studies in Belarus and will encourage funding increase and the corresponding development and expansion of sequencing facilities for molecular surveillance.

The second limitation is that phylogeographic inference of introduction sources can be sensitive to sampling bias and can be affected by the relatively slow accumulation of mutations in SARS-CoV-2 genomes[12,18,39]. In particular, even though no transmission links with Ukraine have been detected, it is likely that such links will emerge when more data from both countries will become available. Thus, SARS-CoV-2 phylogeography analysis should always be treated with a grain of salt, even though the transmission history presented in this study is consistent enough and agrees with the travel records for those cases when they are available. The source inference for Belarus during the early pandemic can actually be more accurate than for some other regions since Belarusian lineages were established after most of their source lineages were already sufficiently diversified. The incorporation of the global travel statistics into the "mugration model" of ref. [27] may also have contributed to the increase in transmission inference accuracy. Finally, for the case of Belarus, even if new data refine the estimation of sources of some lineages, the obtained results are likely reflecting a true trend towards the higher prevalence of regional and neighbor-to-neighbor virus importations.

The third limitation is the sparsity of the SARS-CoV-2 incidence and testing data. In contrast to other countries[52,53], Belarusian COVID-19 statistics are currently reported only for the entire country rather than for specific regions. The reported numbers of tests are not dichotomized into first-time tests and retests, those conducted by state or commercial laboratories, PCR and antibody tests. Furthermore, the sampling for testing is likely incomplete and biased towards individuals with COVID-19 symptoms and their close contacts and, for instance, persons who were tested upon arrival or prior to departure from the country. These issues may result in underestimation of the true number of cases, even though we are employing a method that is supposed to take them into account. For example, if a considerable number of recovered individuals were tested at least twice, then the adjusted proportion of positive tests among those who are getting tested the first time will be higher and, consequently, the estimates of the number of cases will also increase. Furthermore, the aforementioned issues impede the development of stochastic agent-based models that otherwise can be used for high-precision analysis and forecasting. If (or when) more precise data will become available, it can be used to improve the precision and accuracy of our estimates.

In conclusion, this study demonstrates the power of SARS-CoV-2 surveillance using combined genomic and epidemiological

data. Molecular surveillance of SARS-CoV-2 in Belarus is coordinated by the Republican Research and Practical Center for Epidemiology and Microbiology (RRPCEM) represented by several co-authors of this paper. Prior to this study, such surveillance allowed to confirmation the virus importation routes for the first cases and to detect newly emerging viral lineages (including alpha and delta variants) relatively soon after their appearance in neighboring countries. This information has been promptly reported to the government decision-makers. However, in the post-COVID era, advanced molecular surveillance necessary for informed and timely public health interventions should be based on deeper analysis, that resembles or extends the analysis reported here. Thus, for such resource-constrained countries as Belarus, it is vitally important to develop sequencing facilities, detailed statistics, and analytical resources to the level already established in other countries. These facilities and resources should become integral parts of the national mechanism to respond to the emergence, re-emergence, and spread of SARS-CoV-2 and other pathogens.

**Reporting summary**. Further information on research design is available in the Nature Research Reporting Summary linked to this article.

## Data availability
The sequences used in this study are available at GISAID[21]. The acknowledgments tables with accession numbers of all these sequences and names of researchers and laboratories who produced them are available at https://github.com/compbel/COVID-Belarus. Source data for the main figures in the manuscript can also be accessed via that Github repository.

## Code availability
The Matlab scripts, Nextstrain configuration files, BEAST 2 XML files used to perform the described analyses are freely available at https://github.com/compbel/COVID-Belarus[54].

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

## Acknowledgements

A.N. was supported by the GSU Brains and Behavior fellowship. P.S. was supported by the National Institutes of Health grant 1R01EB025022 and by the National Science Foundation grant 2047828.

## Author contributions

A.N. performed a phylogenetic analysis, analyzed genomic data, and wrote the paper. A.E.A. performed a phylogenetic analysis and analyzed genomic data. E.G., K.B., L.V., and A.K. prepared and handled genomic and associated epidemiological data, carried out the primary sequence processing. O.G. analyzed genomic data and wrote the paper. A.K. supervised the incidence data analysis, processed and analyzed incidence data, wrote the paper. P.S. designed and supervised the study, designed and implemented bioinformatics algorithms, performed a phylogenetic analysis, analyzed genomic data, and wrote the paper.

## Competing interests

The authors declare no competing interests.
