## [Peer Review File · Communications Medicine]

Reviewers' comments:

Reviewer #1 (Remarks to the Author):

Nemira and colleagues present on the patterns of genomic epidemiology using SARS-CoV-2 genomes sampled from Belarus. This manuscript is a good example of a collaboration that directly involves researchers in Belarus. This data sovereignty is reassuring to see particularly with rapid public sharing of genomic data.

Although limited in sample number, this manuscript presents an interesting insight onto transmission patterns of SARS-CoV-2 in an under-sampled region. For this reason I recommend acceptance of the manuscript since it nicely fills a gap in the literature about these dynamics. The methods are clear and robust and the article is well-written. I have only a few minor suggestions that might help to improve the paper.

Various public health interventions are described in the introduction. It would help the reader if these were summarised on a figure of the epidemic curve in Belarus with the timings of intervention measures and when these measures were relaxed. It might also be helpful to show the number of genomes that were sampled over the same timeframe to show if these were temporally and proportionally representative.

Please state the number of genomes sequenced at the start of the results section and the proportion of reported cases this represented.

All of the different colours in Figure 2 are confusing. The branches and tips corresponding to different things is difficult to interpret. Instead of showing panel C in this way, could you plot the number or size of transmission lineages that originated in a given region?

Can you explicitly say whether or not genome sequencing of SARS-CoV-2 has played into public health interventions in managing outbreaks in the country?

Reviewer #2 (Remarks to the Author):

This study is a phylogenomic and phylodynamic exploration of COVID-19 in Belarus. The study uses the available genomic data to estimate the number of importations to Belarus and assess the effective reproductive number prior-, and post- implementation of non-pharmaceutical interventions (NPI) during the first wave of COVID infections in Belarus. While such analysis has been done in many settings, especially those with intensive COVID suppression programs, the strength of this manuscript is that Belarus has implemented few NPIs, providing a comparison to both previous studies and an analysis of publicly available data from neighbouring countries, where a broader range of NPIs have been implemented.

The major limitation of this study, as acknowledged by the authors, is the small number of sequences available, only 41 included despite current case counts in excess of 400,000. In light of this:

- Can the authors please include information on how samples were selected for sequencing and, if

these are a convenience sample, any known biases in the sample selection or provide any assessment of the representativeness of these sequences? In what settings (research, public health surveillance etc.) has this work been performed?

- Both the number of importations identified, and the number of intra-country transmission clusters is low. Can the authors please comment on how these estimates may have been impacted by the included sample? I think it important to note these are vast estimations based on the available dataset.

- Belarus has experienced two major waves of COVID infections. It is a shame that the available data did not allow any comparison between first and second wave time periods.

In addition, I found the time periods relevant to each analysis confusing. Understanding would be enhanced if the authors could provide an epidemic curve annotated with the time NPI's were implemented in Belarus and the time frame relevant to each analysis (likely supplementary).

Finally, the authors comment in the discussion they hope that this study will further SARS-CoV-2 genomic epidemiology in Belarus. It would be useful if the authors could comment on any public health engagement with sequencing laboratories and/or data in Belarus.

1 Reply to reviewer’s comments

We thank the reviewers for useful comments. All of them have been addressed in the revised version of the manuscript and/or in our reply. The changes made in reply to the comments are highlighted in red in the marked up version of the manuscript.

1.1 Review 1

Comment 1. *Various public health interventions are described in the introduction. It would help the reader if these were summarised on a figure of the epidemic curve in Belarus with the timings of intervention measures and when these measures were relaxed. It might also be helpful to show the number of genomes that were sampled over the same timeframe to show if these were temporally and proportionally representative.*

Reply. We have created a figure. As suggested by the Reviewer 2, it was added to the Supplement, please see Fig. S2. We also would like to note that the dynamics of the number of lineages, which is the most relevant for the phylodynamic analysis, is depicted on Fig. 1D.

Comment 2. *Please state the number of genomes sequenced at the start of the results section and the proportion of reported cases this represented.*

Reply. These numbers have been added.

Comment 3. *All of the different colours in Figure 2 are confusing. The branches and tips corresponding to different things is difficult to interpret. Instead of showing panel C in this way, could you plot the number or size of transmission lineages that originated in a given region?*

Reply. We agree that the notations on Fig. 2B can be confusing, and we have updated them to make the figure more clear. Now colors encode transmission lineages (as before), and regions are encoded by tip labels of different shapes. We hope that it makes the figure more readable. The numbers of transmission lineages that originated in a given country are depicted on Fig. 1B. We tried to incorporate it into Fig. 2, but the resulting layout turned out to be rather ugly. We also agree that it was not properly linked to Fig. 2, and we have added the corresponding text. Fig. 2C was intended to serve a different purpose and demonstrate that most Belarusian transmission lineages originated from its geographic neighbors (in contrast to many other countries, as discussed in the paper). Therefore we would prefer to keep it, if possible. We agree that its purpose was unclear, and we have added the explicit explanation to the text.

Comment 4. *Can you explicitly say whether or not genome sequencing of SARS-CoV-2 has played into public health interventions in managing outbreaks in the country?*

Reply. The following text was added to the Discussion: "Molecular surveillance of SARS-CoV-2 in Belarus is coordinated by Republican Research and Practical Center for Epidemiology and Microbiology (RRPCEM) represented by several co-authors of this paper. Prior to this study, such surveillance allowed to confirm the virus importation routes for the first cases and to detect newly emerging viral lineages (including alpha and delta variants) relatively soon after their appearance in neighboring countries. This information has been promptly reported to the government decision makers. However, in post-COVID era, advanced molecular surveillance necessary for informed and timely public health interventions should be based on deeper analysis, that resembles or extends the analysis reported here."

1.2 Review 2

Comment 1. *Can the authors please include information on how samples were selected for sequencing and, if these are a convenience sample, any known biases in the sample selection or provide any assessment of the representativeness of these sequences? In what settings (research, public health surveillance etc.) has this work been performed?*

Reply. The following text has been added to the data description in Methods section: "Specimen and their metadata were originally collected by the Republican Research and Practical Center for Epidemiology and Microbiology (RRPCEM, 40 specimen) and Gomel State Medical University (1 specimen) in surveillance settings. The sequenced samples were selected to represent all 7 major regions of Belarus; however, 60% of sequences represent the capital city of Minsk, where the testing facilities were better developed and whose population constitutes 22% of the country's total population. Females constitute 51% of sampled individuals, the age distribution was: ≤ 20 years – 15%, 21 – 40 years – 26%, 41 – 60 years – 49% and ≥ 61 years – 10%."

Comment 2. *Both the number of importations identified, and the number of intra-country transmission clusters is low. Can the authors please comment on how these estimates may have been impacted by the included sample? I think it important to note these are vast estimations based on the available dataset.*

Reply. We agree that the estimated numbers of clusters are lower bounds on the true numbers that are provided by the existing data. To emphasize it, the following text has been added to the Discussion: "First, the analysis revealed the diverse history of transmissions of SARS-CoV-2 into, from and inside the country. It identified 18 introductions within 11 genomic lineages, which is comparable with similar early estimates for other regions [15, 22, 28, 35, 47]. However, for countries with denser sequence sampling these numbers are significantly larger [23, 27, 37]; thus the estimate for Belarus is most likely a lower bound on the real number of introductions. Still, it allows for several important observations."

Comment 3 *Belarus has experienced two major waves of COVID infections. It is a shame that the available data did not allow any comparison between first and second wave time periods.*

Reply. We completely agree with this comment. The following text has been added to the Discussion section: "Furthermore, new data may allow to compare first and second COVID-19 waves, which is especially interesting for understanding the epidemiological effects of social and public health developments during the second half of 2020."

Comment 4 *In addition, I found the time periods relevant to each analysis confusing. Understanding would be enhanced if the authors could provide an epidemic curve annotated with the time NPI's were implemented in Belarus and the time frame relevant to each analysis (likely supplementary).*

Reply. We have added the figure, please see Supplemental Fig. S1.

Comment 5 *Finally, the authors comment in the discussion they hope that this study will further SARS-CoV-2 genomic epidemiology in Belarus. It would be useful if the authors could comment on any public health engagement with sequencing laboratories and/or data in Belarus.*

Reply. The following text was added to the Discussion: "Molecular surveillance of SARS-CoV-2 in Belarus is coordinated by Republican Research and Practical Center

for Epidemiology and Microbiology (RRPCEM) represented by several co-authors of this paper. Prior to this study, such surveillance allowed to confirm the virus importation routes for the first cases and to detect newly emerging viral lineages (including alpha and delta variants) relatively soon after their appearance in neighboring countries. This information has been promptly reported to the government decision makers. However, in post-COVID era, advanced molecular surveillance necessary for informed and timely public health interventions should be based on deeper analysis, that resembles or extends the analysis reported here.”

REVIEWERS' COMMENTS:

Reviewer #1 (Remarks to the Author):

All revisions have been made. Sorry for the delay.

Reviewer #2 (Remarks to the Author):

Thank you to the authors for their attention to the requested changes and congratulations on their interesting manuscript.

Reply to reviewer's comments

We thank the reviewers for their evaluation of our work.

Reviewer #1 (Remarks to the Author):

All revisions have been made. Sorry for the delay.

Reviewer #2 (Remarks to the Author):

Thank you to the authors for their attention to the requested changes and congratulations on their interesting manuscript.